# Seismic Multiple Attenuation in the Continent–Ocean Transition Zone of the Northern South China Sea

**Nan Chen** [1,2] [ID], **Chun-Feng Li** [3,4,*], **Yong-Lin Wen** [3], **Peng Wang** [5], **Xiu-Lian Zhao** [5] and **Xiao-Li Wan** [1]

1. State Key Laboratory of Marine Geology, Tongji University, Shanghai 200092, China
2. Ningbo Regional Railway Investment and Development Co., Ltd., Ningbo 315111, China
3. Institute of Marine Geology and Resources, Zhejiang University, Zhoushan 316021, China
4. Hainan Institute, Zhejiang University, Sanya 572000, China
5. SINOPEC Shanghai Offshore Petroleum Company, Shanghai 200120, China
* Correspondence: cfli@zju.edu.cn

**Abstract:** In this study, we process four new multichannel reflection seismic profiles acquired in 2015 and 2016 in the continent–ocean transition zone (COT) of the northern South China Sea (SCS). We apply a multi-domain, progressive, and seabed-controlled denoising technique and obtain a good denoising effect. Combining velocity analysis in the multi-round time domain and forward modeling, we analyze the types and characteristics of multiples in the study area and formulate an effective demultiple technique to attenuate strong seabed multiples, diffracted multiples from rough seafloor, and other multiples from deep reflectors. The processing results show that the sea surface-related multiple elimination technique predicts the sea surface-related multiples accurately by data convolution, and has a good effect in attenuating seabed multiples. Diffracted multiple attenuation method extracts high-frequency and high-energy diffracted multiples, and suppresses multiples by the energy ratios of multiples to primary events. To attenuate deep multiples, we select predictive deconvolution to attenuate periodic deep multiples after many trials and detailed analysis. The combination of these different techniques in sequence proves to be quite effective in attenuating different seismic multiples in the COT. The imaged crustal structures near the COT often show strong magmatism and/or basement uplifting. The faulted and thinned continental crust adjacent to the COT corresponds to the lowest free-air gravity anomalies. Gravity anomalies often increase from the COT to the oceanic crust. An exception is to the northeast of the SCS, where the relatively wide COT shows very high gravity anomalies, likely induced by mantle upwelling and serpentinization.

**Keywords:** South China Sea; continent–ocean transition; seismic exploration; data processing; multiple attenuation; gravity anomaly





## 1. Introduction

The rift-to-drift transition and accretion of oceanic crust are important to the plate tectonic cycle. The South China Sea (SCS) is one of the largest marginal seas of the western Pacific. It underwent nearly a complex cycle from continental breakup to seafloor spreading and subduction [1], and therefore represents an ideal place to study the lithospheric breakup and oceanic accretion. The tectonic deformation characteristics in the continent–ocean transition zone (COT) are key to understanding the rift-to-drift process [2].

Offshore oil accounts for ~34% of the world's total oil resources [3], and the exploration trend is now in transition from deep water (400–1500 m) to ultra-deep water plays (>1500 m) [4]. Reflection seismic exploration in the COT of the SCS is of great significance to deep-water gas and oil exploration and to the study of lithospheric structures and tectonic rifting of the SCS. The transition zone has been drilled recently by three International Ocean Discovery Program (IODP) expeditions (#349, 367, and 368/368X) [5,6].

In the COT of the northern SCS, low-velocity Cenozoic strata cover sets of high-velocity carbonate and igneous rocks and/or Mesozoic sedimentary rocks, generating quite strong



impedance contrasts [7,8]. These contrasts have a strong shielding effect on seismic wave propagation and generate strong sea surface-related multiples. Diffractions as a result of variation of seabed topography, coupled with the vibration, free surface multiples, and refraction multiples, cause a variety of strong energy disturbances on primary events, loss of frequency component of the signals, and reduction of seismic resolution. Multiple attenuation is critical in marine seismic data processing to improve the quality of the seismic data.

At present, multiple attenuation methods can be classified into two types, wave equation method and filter method [9–11]. No single method can be applicable to all types of multiple or completely suppress each type of multiple. In this research, we characterize the multiples in the COT of the northern SCS and propose a combination of demultiple techniques to improve seismic resolution and signal-to-noise ratio.

## 2. Materials and Methods

### 2.1. Geological Setting and Data

The SCS is located in the conjunction zone of the Pacific, Eurasian, and India-Australia plates, with an area of about 3.5 million km$^2$. The South China continental margin entered a stage of rifting and extension during the Late Cretaceous (~100 Ma) [12]. Seafloor spreading in the SCS basin continued from the late Oligocene to the middle Miocene [1,13,14] and formed three subbasins (Northwest, Southwest, and East subbasins) (Figure 1). Extensive magmatic activities occurred during and after the seafloor spreading stage.

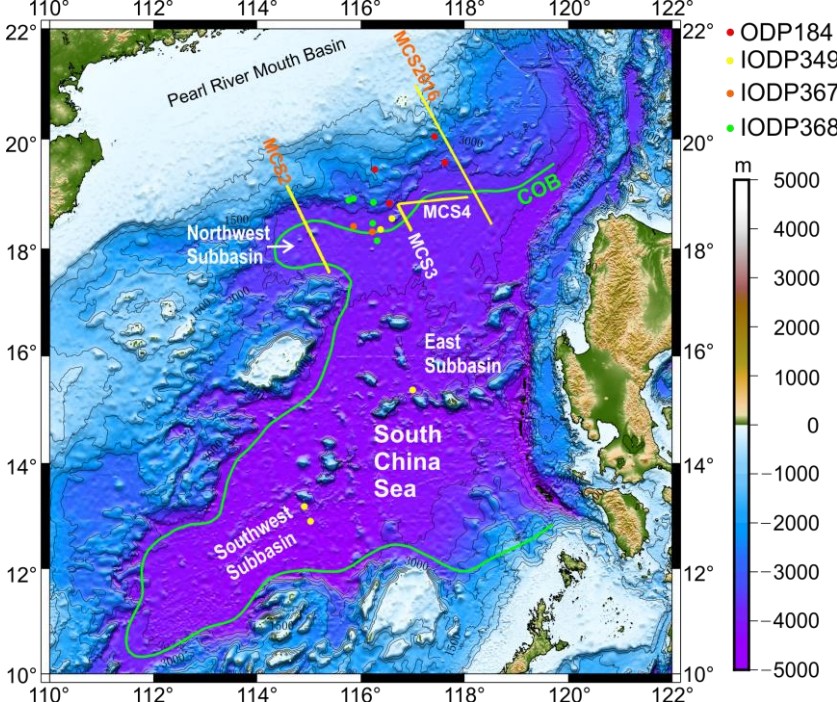

**Figure 1.** Bathymetry and tectonic units of the South China Sea. The green line marks the continent-ocean boundary (COB) [15]. MCS2, MCS3, MCS4, and MCS2016 are the seismic profiles used in this research. The red dots show the 5 drill sites of Ocean Drilling Program (ODP) Leg 184, the yellow dots show the 5 drill sites of Integrated Ocean Discovery Program (IODP) Expedition 349, and the brown and green dots are the drill sites of IODP Expeditions 367 and 368, respectively.

The SCS has a broad prospect in gas and oil exploration and production. Reservoirs in the petroliferous sedimentary basins in the northern margin of the SCS are mainly of clastic rocks, and different hydrocarbon traps are well developed. The COT separates the continental crust from the distinct oceanic crust (Figures 1 and 2) [16–18]. In the continental margin of the northern SCS, magmatic rocks in the upper crust and high-velocity layers in

the lower crust were developed [19–21]. Deep reflection seismic profiles show that basin boundary faults can cut into the lower crust [22].

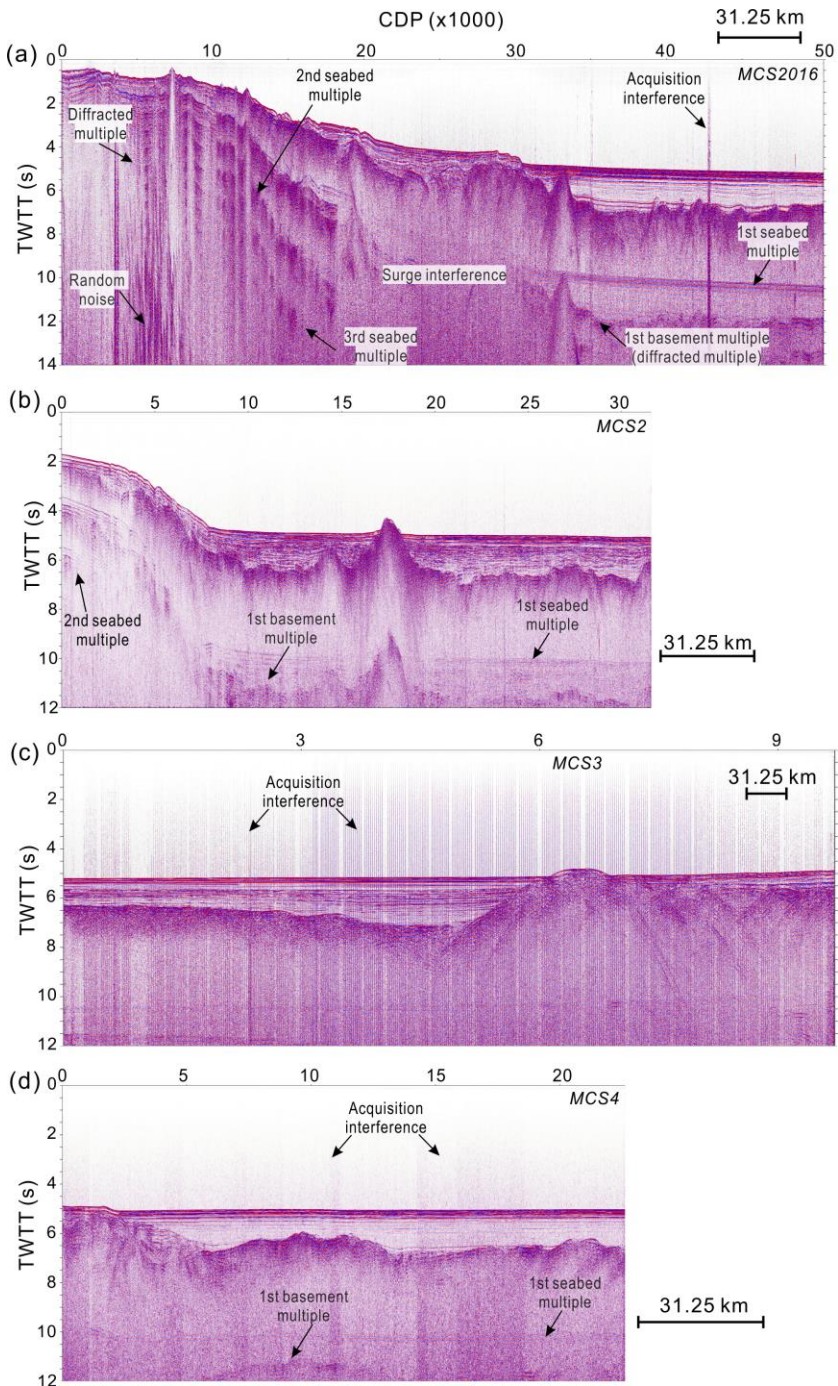

**Figure 2.** Raw stacked sections of lines MCS2016 (**a**), MCS2 (**b**), MCS3 (**c**), and MCS4 (**d**). Different types of multiples (seabed multiples, diffracted multiples, etc.) and noise (random noise, surge noise, and acquisition interference) are marked with arrows.

The new seismic data of this research came from joint geophysical surveys funded by the National Natural Science Foundation of China in 2015 and 2016. The aims of these surveys were to investigate deep structures and rifting dynamics in the continental margin of the northern SCS and conduct marine geophysical surveys. We deployed the R/V "Shiyan 2" of the South China Sea Institute of Oceanology, Chinese Academy of Sciences, to acquire multi-channel reflection seismic (MCS), single-channel reflection seismic (SCS),

ocean bottom seismic (OBS), and ocean bottom electromagnetic (OBEM) data (Figure 1; Table 1). We carried out systematic processing including surge and noise attenuation, several rounds of time domain velocity analysis, different types of multiple attenuation, and pre-stack time migration.

**Table 1.** Reflection seismic data acquisition parameters.

| Survey lines | MCS2016 | MCS2 | MCS3 | MCS4 |
|---|---|---|---|---|
| Cable length (m) | 1500 | 1500 | 1500 | 1500 |
| Channel number | 120 | 120 | 120 | 120 |
| Trace interval (m) | 12.5 | 12.5 | 12.5 | 12.5 |
| Minimum offset (m) | 150 | 145 | 145 | 145 |
| Maximum offset (m) | 1638 | 1633 | 1633 | 1633 |
| Shot interval (m) | 200 | 50 | 50 | 50 |
| CDP interval (m) | 6.25 | 6.25 | 6.25 | 6.25 |
| Sampling interval (ms) | 2 | 2 | 2 | 2 |
| Record length (s) | 14 | 12 | 12 | 12 |
| Cable depth (mbsl) | 12 | 12 | 12 | 12 |
| Airgun depth (m) | 10 | 10 | 10 | 10 |
| Length of survey line (km) | 321 | 200 | 63 | 142 |
| Total number of shots | 1572 | 3944 | 1200 | 2814 |
| Total number of CDP | 50,392 | 31,664 | 9712 | 22,624 |
| Fold | 4 | 15 | 15 | 15 |
| Year of acquisition | 2016 | 2015 | 2015 | 2015 |

MCS2016 data source: Joint geophysical surveys funded by National Natural Science Foundation of China in 2016. MCS2, MCS3, and MCS4 data source: Joint geophysical surveys funded by National Natural Science Foundation of China in 2015.

*2.2. Reflection Seismic Data Processing*

Multiples are clearly evident on the raw stacked sections of MCS2016, MCS2, MCS3, and MCS4 (Figure 2a–d). Strong first-order seabed and basement multiples are easily recognizable. Second- and even third-order multiples that interfere with primary events can also be seen on seismic profiles (Figure 2a,b). There are diffracted multiples in the area of rough seabed (Figure 2b). Low-frequency surge noise and random noise are also common, including artificial interference in acquisition (Figure 2a,c,d).

Considering the noise from the study area, we formulate the following strategy of noise attenuation with velocity analysis in the multi-round time domain (Figure 3).

1. Prestack noise attenuation: Suppress surge noise, random noise, and strong seabed multiples by a multi-domain noise suppression technique.
2. Multiple attenuation: Select a combination of effective demultiple techniques to attenuate strong seabed and diffracted multiples, including sea surface-related multiple attenuation technique, diffracted multiple attenuation, and predictive deconvolution.
3. True amplitude processing: Perform relative amplitude retention and amplitude quality control at each step of the processing.
4. Detailed velocity analysis: Build a reasonable initial migration velocity model by detailed velocity analysis in the multi-round time domain and pre-stack time migration (PSTM).

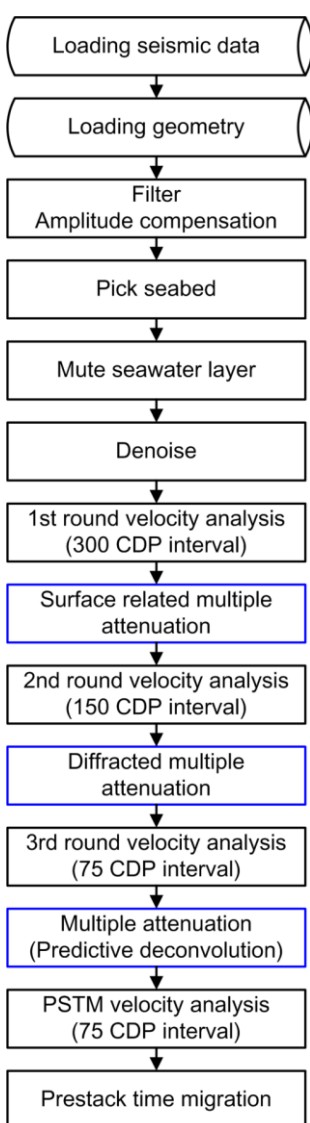

**Figure 3.** Seismic data processing flow. All the seismic data were processed with CGG Geoclusteur system.

## 3. Results

### 3.1. Surge and Random Noise Attenuation

Common noise in marine seismic data can be divided into two categories: low-frequency surge noise and random noise. Surge noise appears during data acquisition in a rough sea condition and/or if the cable depth is shallow. Surge noise shows in low-frequency vertical bands on shot gathers [23]. The random noise can be further divided into two subcategories: environmental noise including the noise from other scientific research vessels [24,25], and the regular or irregular residual noise in processing, both of which must be attenuated before velocity analysis. We can see the surge noise and random noise from the raw shot gathers of MCS2 and MCS2016 (Figure 4), and the frequency of the surge noise is quite low (0–10 Hz) by analyzing the spectrum of MCS2 (Figure 5).

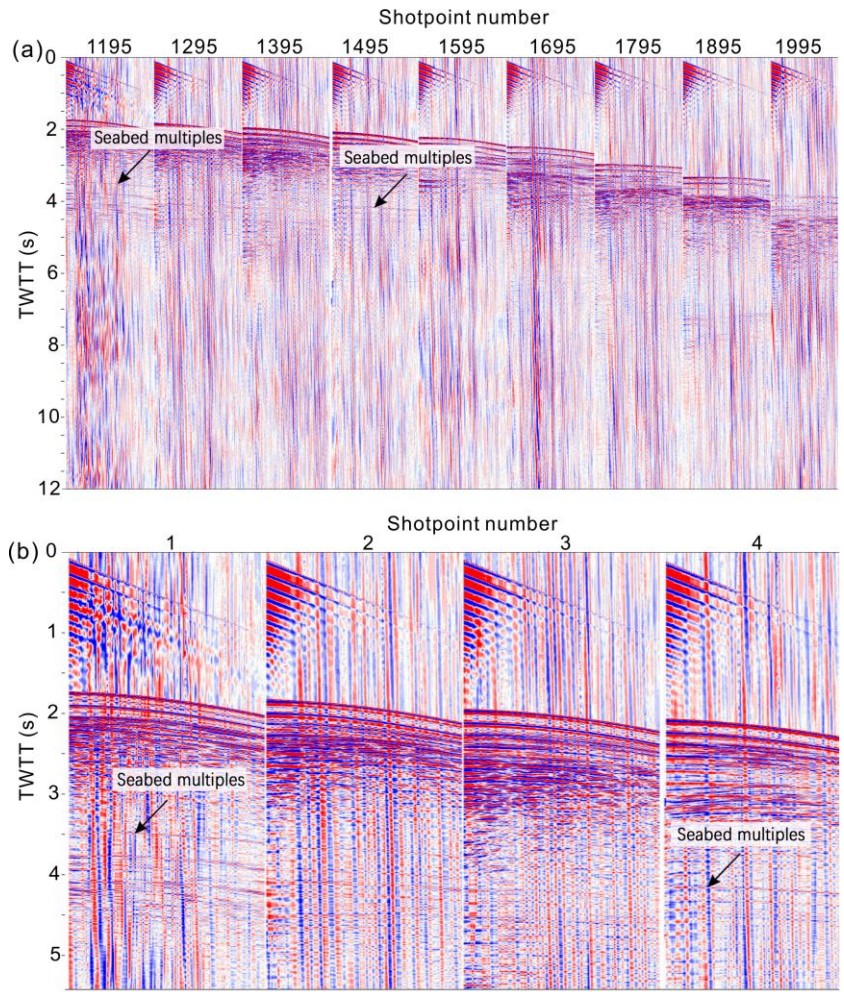

**Figure 4.** Examples of raw shot gathers of lines MCS2 (**a**) and MCS2016 (**b**) showing surge noise and random noise, and seabed multiples.

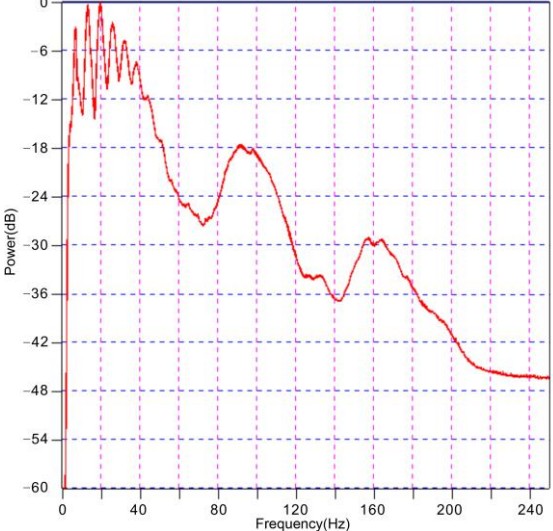

**Figure 5.** Spectral analysis of line MCS2.

The traces are transformed into the frequency domain, and the random noises are attenuated by a signal preservation f-x projection filter, which separates the predictable signal in the f-x domain from non-predictable noise for all component frequencies. The signal is

preserved after filtering, while optimizing the attenuation of random noise. Surge noises are attenuated by the frequency-dependent noise attenuation technique, which attenuates high-amplitude surge noises in decomposed frequency bands and uses frequency-dependent and time-variant threshold values of amplitude samples. Surge noises are detected and attenuated in different time windows and different frequency ranges, to prevent signals from being identified as noise. These techniques are carried out on different traces repeatedly in order to attenuate noise better.

Stacked sections of MCS3 and MCS2016 before and after noise attenuation (Figure 6) show that the low-frequency surge noise, random noise, and the noise from data acquisition are well attenuated and the effective energy is enhanced.

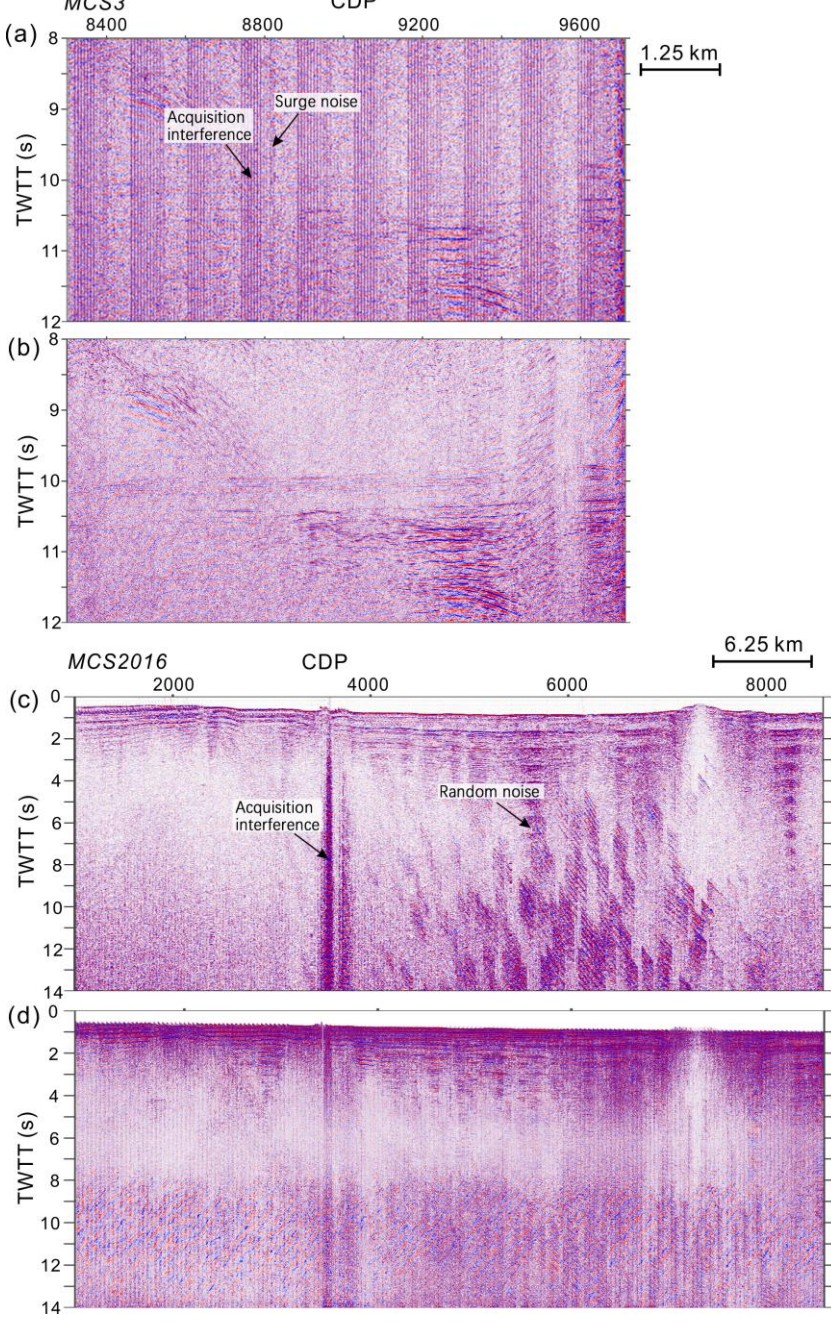

**Figure 6.** Stacked sections before and after denoise of line MCS3 (**a**,**b**) and line MCS2016 (**c**,**d**). Different types of noise are marked with arrows.

We design a combination of different techniques to attenuate sea surface-related multiple, diffracted multiple, and deep multiple, and the results of the velocity analysis in the time domain during the four rounds can be all included.

### 3.2. Multiple Attenuation

3.2.1. Sea Surface Related Multiple Attenuation

In recent years, sea surface-related multiple attenuation technique has become quite successful in multiple attenuation of marine reflection seismic data. It can be divided into two steps, multiple prediction of raw seismic data and multiple subtraction; the predicted multiple must be matched (in amplitude and phase) with the true multiple by least-squares subtraction [26]. The sea surface is the shallowest downward reflection interface that causes multiple [27]. Seabed multiple takes up most of the energy. Seabed multiple is centered around the 10 s in two-way travel time (TWTT) in our seismic profiles (Figure 2a,b,d). Figure 4a shows a strong periodicity of multiple in near-offset channels in the shot gathers, but no obvious periodicity of multiple at large offsets.

Sea surface-related multiple attenuation cannot completely eliminate multiples and there is always residual multiple energy on the stacked sections (e.g., Figure 7b,d), especially for our data of very low fold (4 or 15; Table 1).

The biggest advantage of this method is that it can predict all sea surface-related multiples from the data only without any a priori information of underground medium [28]. The precision of multiple prediction in large offsets by this technique is usually limited by the boundary problem of convolution for the seismic data, which are divided into blocks (different time windows) and processed in batches (multiple prediction first, and then multiple attenuation).

A multiple model is calculated by match-filtering in each time window, and the main matching parameters are filter length, calculation window, and adjacent trace numbers, which will directly influence processing results [29]. Multiple attenuation is not effective if the filter length is too short, but primary events will be attenuated if the filter length is too long. A filter length of 40 ms is selected after many trials and detailed analysis. The purpose of time windowing is to protect the adjacent primary events when attenuating multiples. The stacked seismic sections are divided into two time windows; the first is above the seabed multiple and the second is the remainder of the stacked sections, which includes sea bed multiples. Multiple attenuation is done in the second time window only. If the adjacent trace number is too large and the distant traces are matched, the demultiple effect will also drop. So, an adjacent trace number of 5 is selected. The residual energy of seismic data should be minimum after multiple attenuation, and we apply adaptive subtraction according to this principle with a seabed control method to protect the primary events above the depth of twice the seabed depth in TWTT. Stacked sections of MCS2 and MCS4 before and after sea surface-related multiple attenuation show effective seabed multiple attenuation while keeping the primary events (Figure 7).

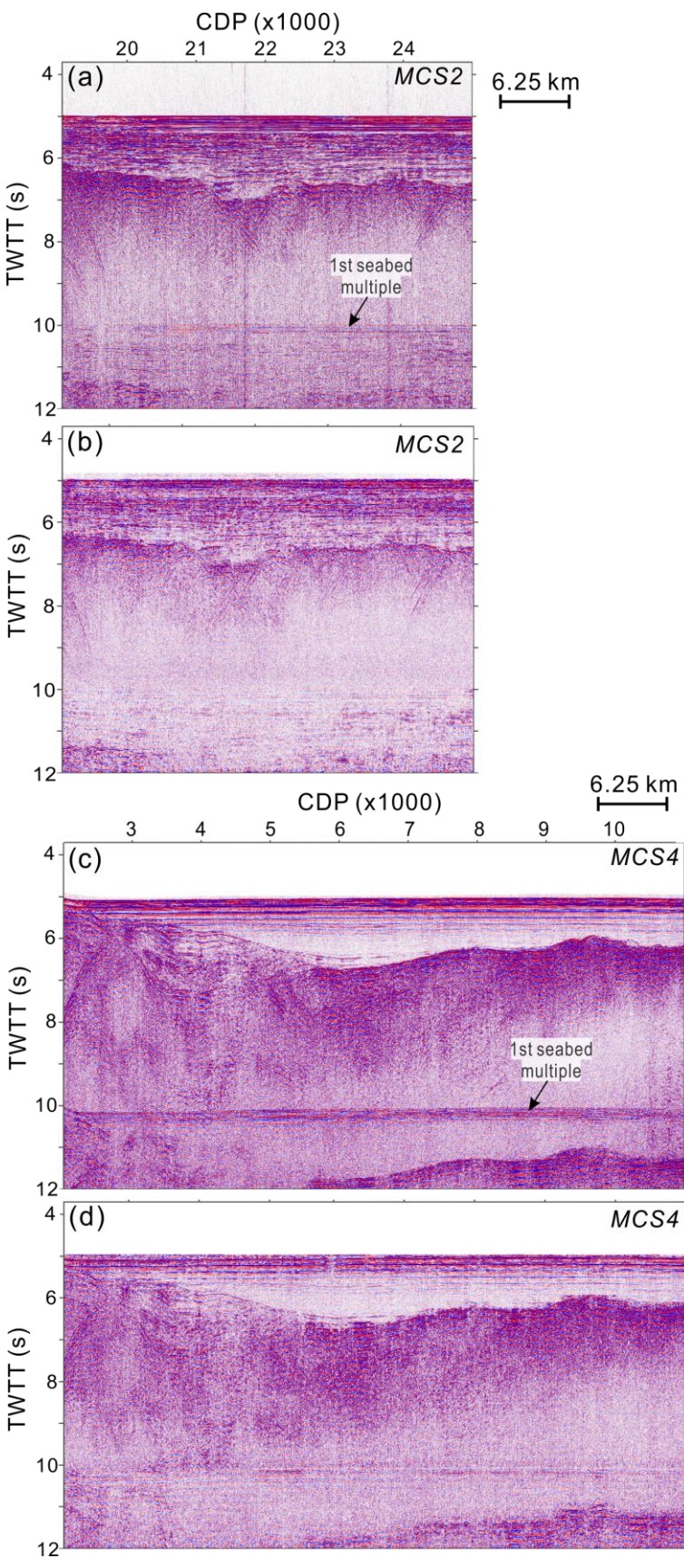

**Figure 7.** Stacked sections before and after sea surface-related multiple attenuation of line MCS2 (**a**,**b**) and line MCS4 (**c**,**d**). Sea surface-related multiples are marked with arrows.

### 3.2.2. Diffracted Multiple Attenuation

It is a single trace technique designed to attenuate diffracted multiples. Multiple suppression is achieved by limiting sample amplitudes within a defined frequency bandwidth. It is usually applied to CMP gathers after conventional demultiple processes to remove diffracted multiple. The method works when there is a difference in the frequency and amplitude between primary and multiple events.

Diffracted multiple attenuation is a key but difficult point. Strong diffracted multiples mix with the primary signal (Figure 2a) and are common in areas with a rough seabed. Diffracted multiples show large randomness in shot gathers for their large variation of propagation paths due to the uncertainty of reflection points [30], and they interfere with primary events. Based on the difference between the primary wave and multiple, we attenuate diffracted multiples by the denoising method from frequency splitting [31]. This method usually attenuates residual diffracted multiples after conventional multiple attenuation and has good effects on multiples of different frequencies and amplitudes [32]. At first, we tried a filter to attenuate diffracted multiples on the whole stack sections based on the fact that the diffracted multiple has higher energy and frequency than primary reflections. We found that the primary events were also attenuated at the same time and the result was not satisfactory. We then attenuated diffracted multiples by a horizon-controlling method, which avoids the area without diffracted multiple, and extracted high-frequency and high-energy diffracted multiples (Figure 8).

### 3.2.3. Deep Multiple Attenuation

Most of the multiple energy is attenuated after sea surface-related multiple attenuation and diffracted multiple attenuation, but still, some of the periodic deep multiples remain (Figure 9).

We found that high-resolution Radon transform and Tau-p transform are not applicable to our low-fold data; the fold of MCS2, MCS3, and MCS4 is 15 and the fold of MCS2016 is 4 (Table 1). The low fold requires heavy interpolation and iterations, which introduce aliasing and lower the computation efficiency [33,34]. After many trials, we selected predictive deconvolution, which works well in attenuating periodic deep multiple [35].

Predictive length and operator length will largely determine the multiple attenuation effect and computation efficiency in predictive deconvolution. Predictive length is the zero-offset moveout of multiples and is related to the period of the multiple. Distant traces with autocorrelation of multiple energy cannot be included if the predictive length is long, and more multiple energy will be retained if the predictive length is closer to distant traces with autocorrelation of sub-maximum [36]. The predictive length in this research is 35 ms. Operator length is the length of the prediction filter operator. Multiples will not be well attenuated and false energy could emerge if the operator length is too short, but heavy computation will be required if the operator length is too long [37]. The operator length in this research is chosen to be 280 ms.

Stacked sections of MCS2 and MCS4 before and after periodic deep multiple attenuation show a good effect of attenuating periodic deep multiple by predictive deconvolution (Figure 10).

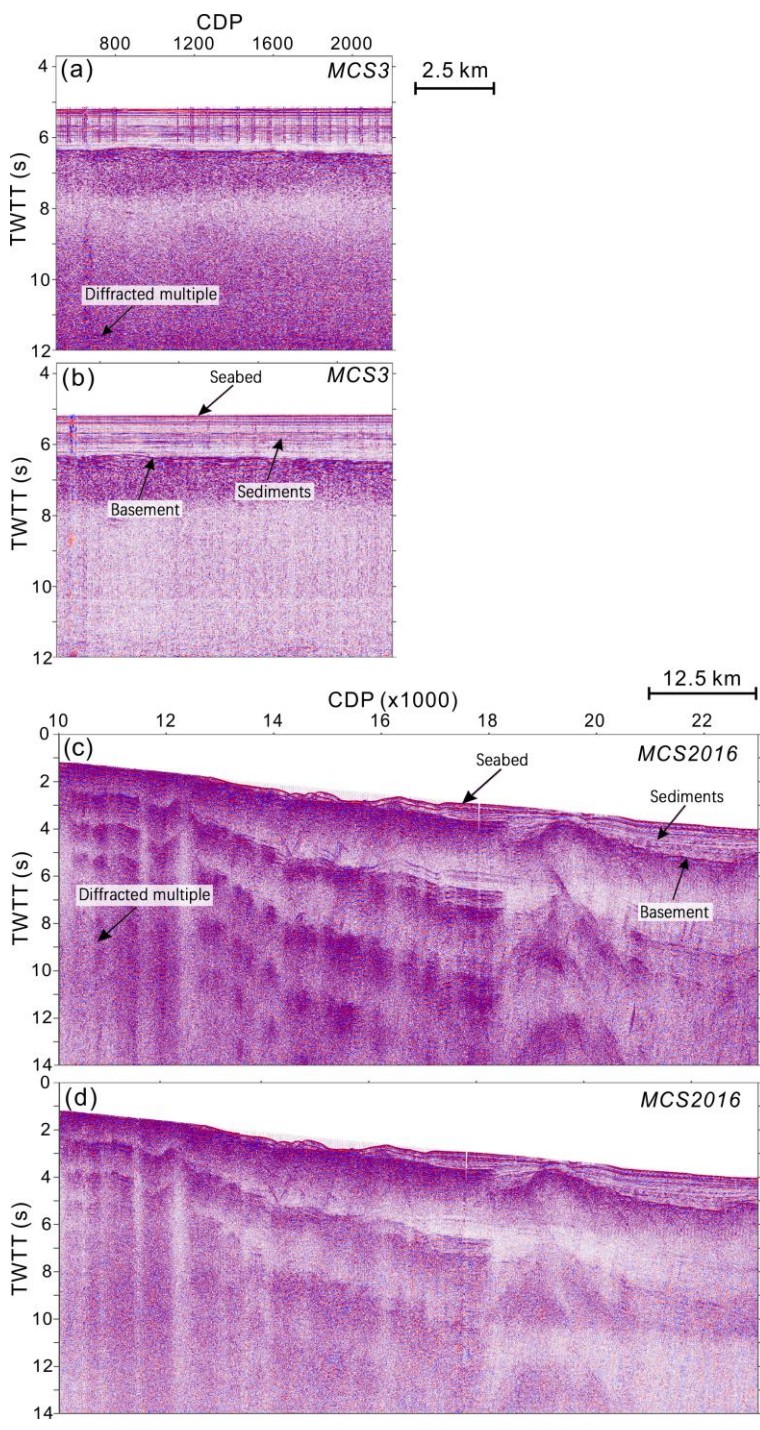

**Figure 8.** Stacked sections before and after diffracted multiple attenuation of line MCS3 (**a**,**b**) and line MCS2016 (**c**,**d**). Diffracted multiples and some geological features are marked with arrows.

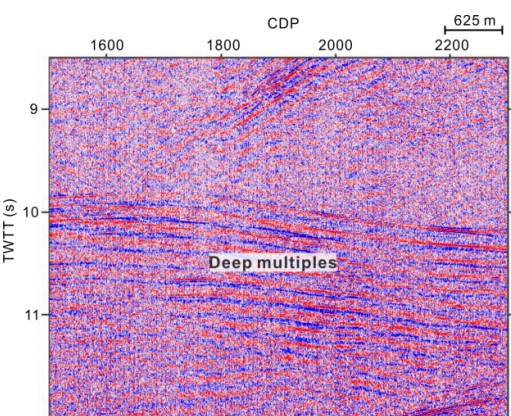

**Figure 9.** Stacked section showing deep multiples of line MCS4.

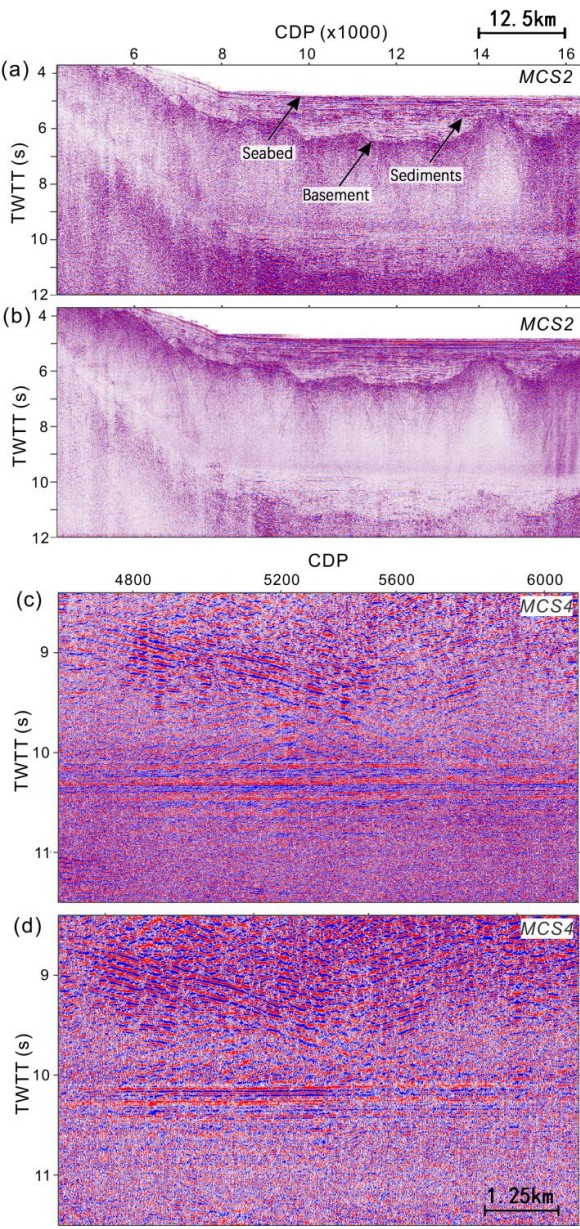

**Figure 10.** Stacked sections before and after predictive deconvolution of line MCS2 (**a**,**b**) and line MCS4 (**c**,**d**). Some geological features are marked with arrows.

### 3.3. Pre-Stack Time Migration

After multiple attenuation and velocity analysis, Kirchhoff pre-stack time migration is applied in the processing for its high calculation efficiency and simple principle, particularly in the area with little variation of lateral velocity [38].

Stacked sections of MCS4 and MCS2 after Kirchhoff pre-stack time migration show a greatly increased signal-to-noise ratio (Figure 11). Primary events are more obvious, and horizons and faults are clearer and more accurate.

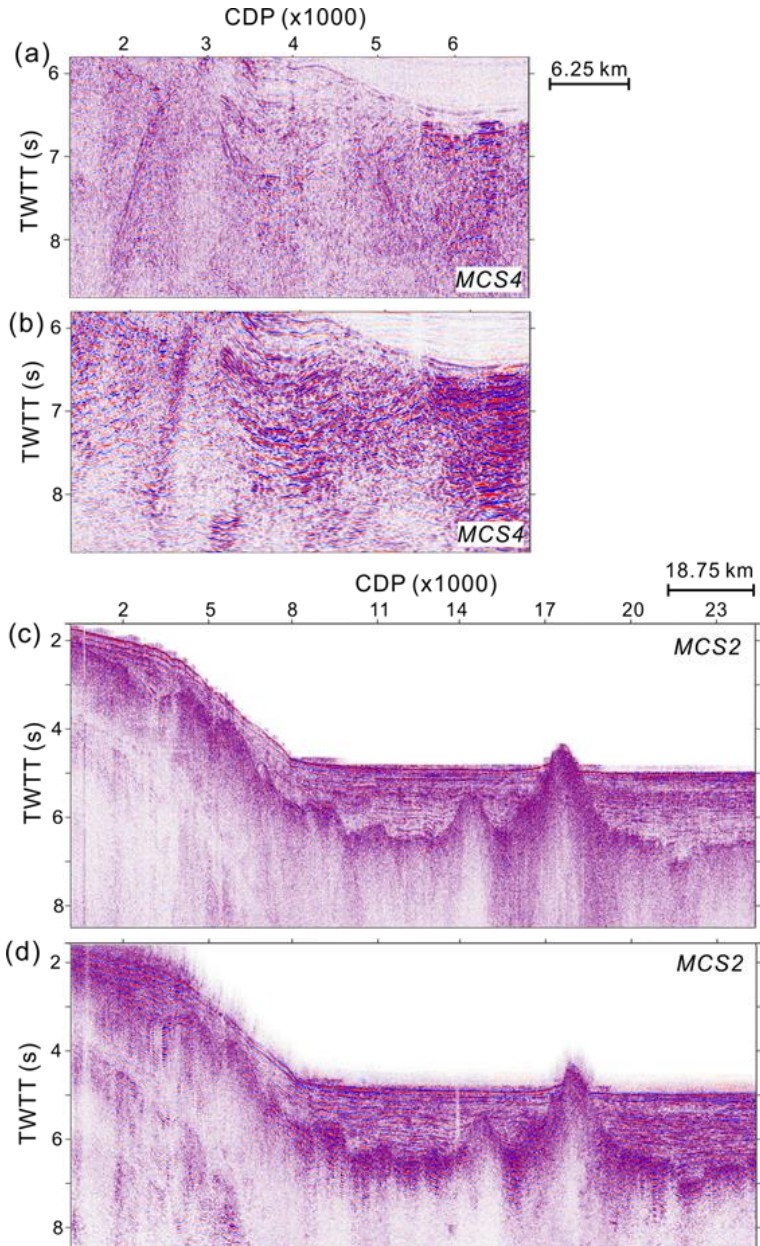

**Figure 11.** Stacked sections before and after Kirchhoff pre-stack time migration of line MCS4 (**a**,**b**) and line MCS2 (**c**,**d**).

### 4. Discussion of Structures in the Continent–Ocean Transition Zone

In seismic profile MCS2016, the extremely thinned continental domain is mainly characterized by normal faults and post-rift volcanoes (100–220 km in Figure 12), and some basement normal faults form graben-like structures overlain by thick sediments. The extremely thinned continental domain also corresponds to a zone of low gravity anomalies,

indicating a relatively low density of the thick sediments in the depression (Figure 12). The continent–ocean transition zone (COT) of about 35 km wide shows normal faulting, magmatic edifices, and interestingly high gravity anomalies, indicating strong mantle uplifting here. The developed normal faults may allow water to penetrate down to the uppermost mantle and cause partial serpentinization, as indicated by high seismic velocity anomalies revealed by an OBS survey along the same transect [21]. Noticeable magnetic anomaly belts already develop within the COT (Figure 12).

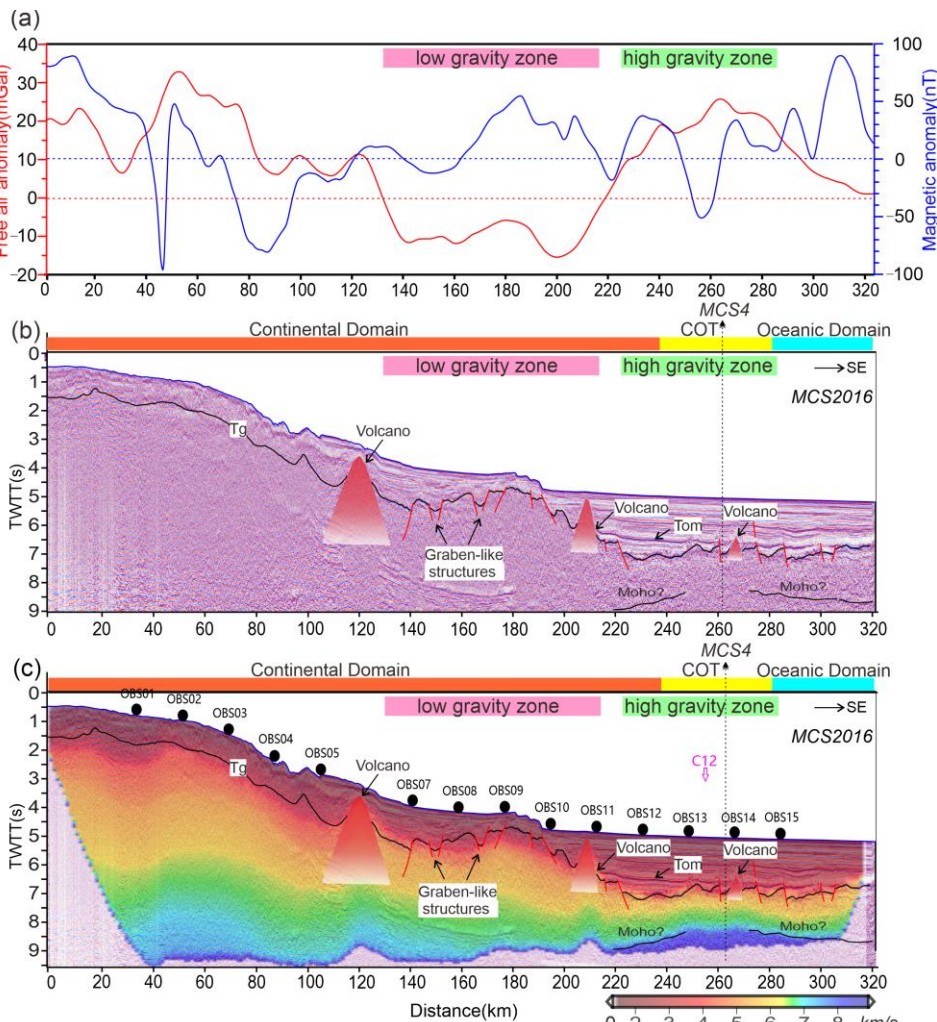

**Figure 12.** (**a**) Free-air gravity and magnetic anomalies along profile MCS2016. (**b**) Seismic interpretation of line MCS2016 (after Kirchhoff pre-stack time migration). (**c**) Velocity structure of OBS survey along the same transect [21]. Tg—basement; Tom—Oligocene–Miocene boundary. Intersection with profile MSC4 is marked. Red lines are interpreted as faults or graben-like structures. Red triangles are interpreted as volcanoes. The location of magnetic lineation C12 [14] is marked in a pink arrow.

The east–west trending seismic profile MCS4 is located mostly within the COT, and in its eastern end profile, MCS4 almost intersects with MSC2016 (Figures 1 and 13). The attenuated continental domain is again characterized by low gravity anomalies and has a fluctuating basement with normal faulting. A thick faulted and tilted syn-rifting pre-Miocene sequence can be seen within the thinned continental domain, and it terminates towards the COT. The COT shows high-amplitude reflections from the top basement, quite different from the continental domain. Gravity anomalies within the COT are mostly positive.

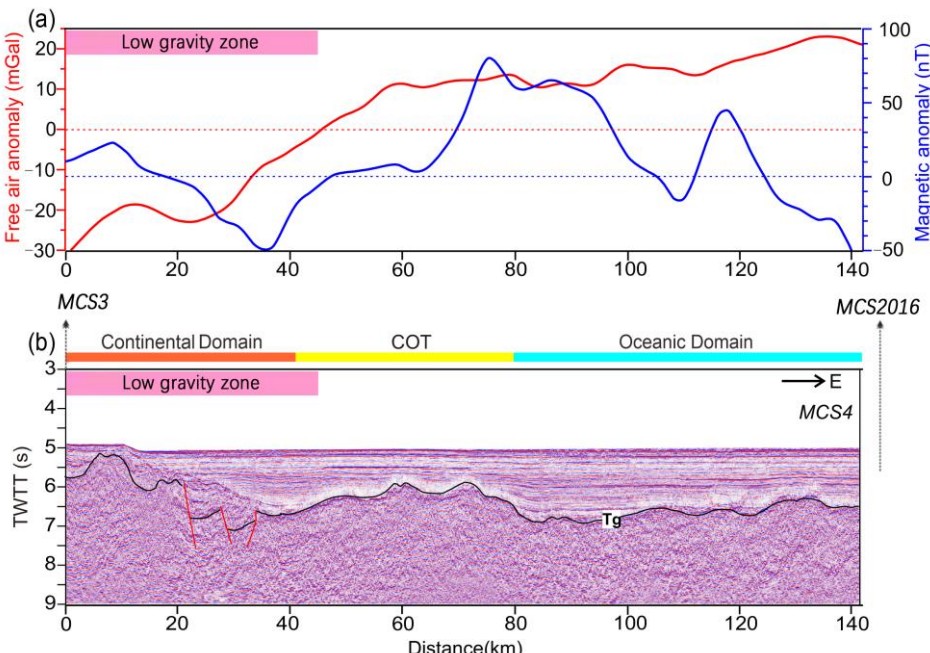

**Figure 13.** (**a**) Free-air gravity and magnetic anomalies along profile MCS4. (**b**) Seismic interpretation of line MCS4 (after Kirchhoff pre-stack time migration). Tg—basement. Intersections between profiles MSC3 and MCS2016 are marked. Red lines are interpreted as faults.

Seismic profile MCS3 intersects at its northwestern end with MCS4 (Figures 1, 13 and 14). As seen in other profiles, the continental domain shows a faulted and undulating basement and very low gravity anomalies. The COT and oceanic crust show high-amplitude reflections from the top basement without large faults. Compared to profile MCS2016, the COT along profile MCS3 appears narrower, and does not show positive gravity anomalies. From the COT southeastwards to the oceanic crust, the basement shallows up, corresponding to increasing free-air gravity anomaly. Again, noticeable magnetic anomaly belts can be observed within the COT (Figure 14).

Further west, seismic profile MCS2 across the Northwest Subbasin shows faulted and thinned lower continental slopes with very low gravity anomalies (Figure 15). The data quality is not ideal due to the low fold number of 15, but the pre-Miocene syn-rifting sequence can be identified with the low gravity zone. Post-breakup deep marine sediments deposited onlap to the syn-rifting sequence. The COT shows a discontinuity in the Moho reflections [15] and is characterized by either intruded volcanoes or basement uplifting (Figure 15). From the COT to the oceanic lithosphere, the gravity anomaly increases steadily.

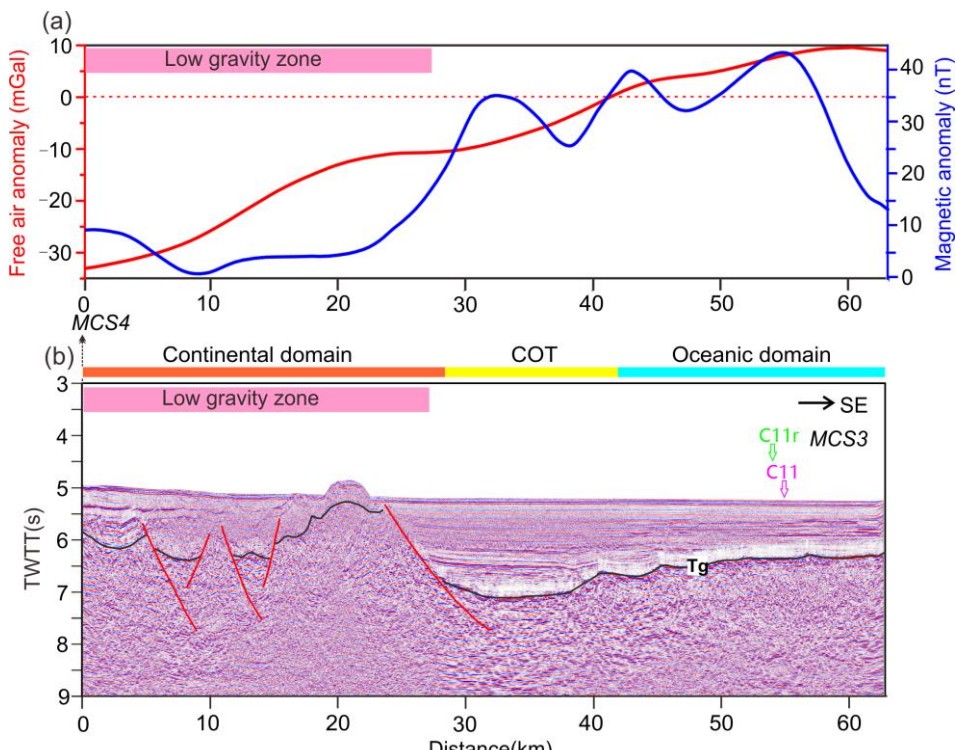

**Figure 14.** (**a**) Free-air gravity and magnetic anomalies along profile MCS3. (**b**) Seismic interpretation of line MCS3 (after Kirchhoff pre-stack time migration). Tg—basement. Intersection between profiles MSC4 is marked. Red lines are interpreted as faults. The locations of magnetic lineations C11 and C11r are marked in pink [14] and green arrows [1].

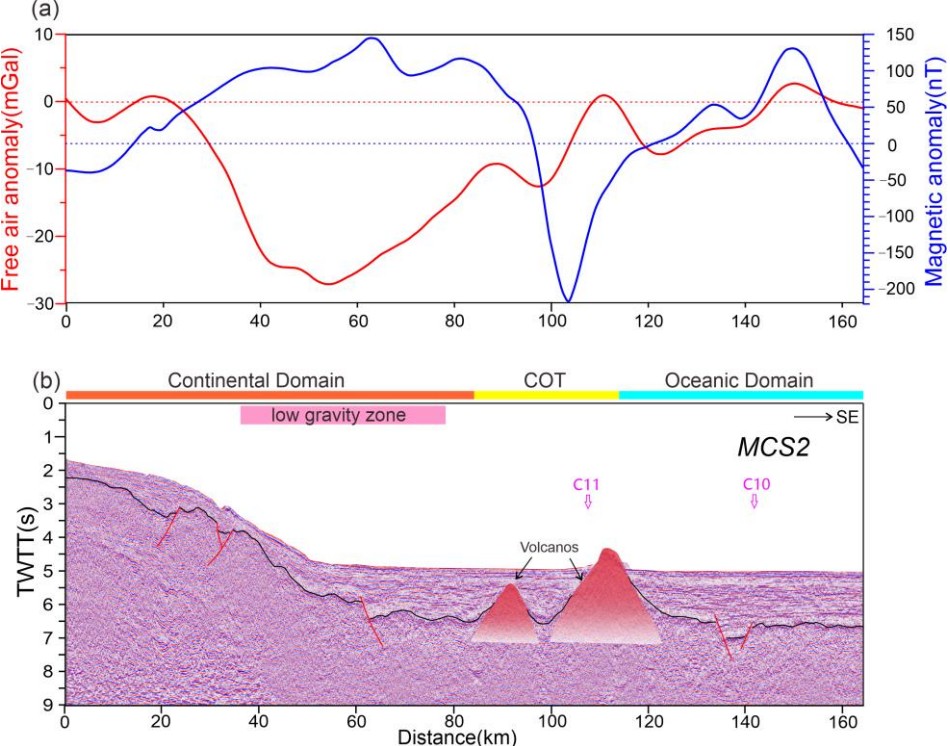

**Figure 15.** (**a**) Free-air gravity and magnetic anomalies along profile MCS2. (**b**) Seismic interpretation of line MCS2 (after Kirchhoff pre-stack time migration). Tg—basement. Red lines are interpreted as faults. Red triangles are interpreted as volcanoes. The locations of magnetic lineations C11 and C10 are marked in pink arrows [14].

## 5. Conclusions

In this study, we analyze and characterize the types of multiple in the continent–ocean transition zone (COT) of the northern South China Sea (SCS) and apply different multiple attenuation methods.

Multistage sea surface-related multiples with strong energy are commonly found, which are periodic in near-offset traces. The diffracted multiples are commonly found in areas with complex submarine topography and large variations in the water depth. The multi-domain, progressive, and seabed-controlled denoising technique is effective in attenuating the surge and random noise in this research. In addition, an effective multiple attenuation scheme is established, and we apply sea surface-related multiple attenuation, diffracted multiple attenuation, and predictive deconvolution in sequence in attenuating different multiples. Sea surface-related multiple attenuation is good at attenuating seabed multiples while keeping the primary events, and predictive deconvolution can attenuate deep periodic multiples, especially in our case of low-fold seismic data. This combination of demultiple techniques may also be applicable to seismic data acquired from other continent–ocean transition zones and continental margins, especially if the seismic data are of low fold.

From interpretations of newly processed seismic profiles, we conclude that the extremely thinned and faulted continental crust adjacent to the COT has the lowest free-air gravity anomalies in the study area, and the gravity anomalies often increase steadily from the transitional crust to the oceanic crust. To the northeast margin of the SCS, however, the COT appears wider and shows very high gravity anomalies along profile MCS2016, which indicate possible upper mantle upwelling and serpentinization. The imaged crustal structures near the COT often show strong magmatism and/or basement uplifting and large lateral heterogeneity.

**Author Contributions:** Conceptualization, N.C. and C.-F.L.; methodology, N.C., P.W.; software, N.C., Y.-L.W., P.W. and X.-L.Z.; validation, C.-F.L.; formal analysis, Y.-L.W., P.W. and X.-L.W.; investigation, N.C.; resources, C.-F.L.; data curation, C.-F.L.; writing—original draft preparation, N.C.; writing—review and editing, C.-F.L.; visualization, N.C.; supervision, C.-F.L.; project administration, C.-F.L.; funding acquisition, C.-F.L. All authors have read and agreed to the published version of the manuscript.

**Funding:** This research was funded by National Natural Science Foundation of China (91858213; 42176055).

**Institutional Review Board Statement:** Not applicable.

**Informed Consent Statement:** Not applicable.

**Data Availability Statement:** Not applicable.

**Acknowledgments:** The data acquisition using R/V Shiyan-2 of the South China Sea Institute of Oceanology (SCSIO) was sponsored by the Oceanographic Research Vessel Sharing Plan of the National Natural Science Foundation of China. We thank all researchers and sailors on Shiyan-2 for collecting the MCS data, which are available at request. We thank Jiubing Chen for his help in data processing, and Xuelin Qiu, Minghui Zhao, Haibo Huang, Enyuan He, and Jiazheng Zhang at SCSIO for their assistance in data acquisition. Comments from anonymous reviewers are much appreciated.

**Conflicts of Interest:** The authors declare no conflict of interest.

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
