# Peer review of "Seismic Multiple Attenuation in the Continent–Ocean Transition Zone of the Northern South China Sea"

_jmse, doi:10.3390/jmse11010227_

Round 1
Reviewer 1 Report
Review of: Seismic multiple attenuation in the continent-ocean transition zone of
the northern South China Sea
This paper applies a number of established techniques for removal (or mitigating the impact) of unwanted reflected signals, multiples, in seismic profiling data. The paper is what some would refer to as a case study of application of known analysis techniques to acquire new knowledge of a specific ocean setting. Aside from the demonstration of techniques, the research strength of the paper is the combined use of gravity and seismic data to characterize the physical aspects of the coastal ocean transition zone in the South China Sea. My impression is that there is some merit in both the description of the analysis technique and the interpretation assisted by the different types of data. However, the text does not provide sufficient guidance for readers to judge whether the different techniques for mitigating the impact of the various types of multiples. And there is not any real attempt to suggest reasons for the different gravity anomalies, given the additional information provided by the seismic images of the subsea structure.
I have made some suggestions that will improve the clarity of the message and assist readers in judging the effectiveness of the multiple removal techniques that are applied. These are listed below. The authors should also make some attempt to suggest reasons for the different gravity anomalies that are consistent with the new knowledge of the sub sea structure from the seismic data. There are also many other suggestions for correcting typos and grammar. There are too many to list separately, so they are included in the annotated version of the text.
Major critical remarks.
1. Throughout the text, I have indicated instances where ‘surface’ should be replaced by ‘sea surface’. The reason for this is that there are many surfaces, ie interfaces, in the marine environment that could reflect seismic data. The sea surface is the one responsible for generating the sea bed multiple. For technical correctness, the interface should be designated as ‘sea surface’.
2. Section 3. In many instances in the text in this section, the authors assert that various features are evident in the seismic sections shown in the corresponding figures. This reviewer had difficulty finding most of them in the figures. I suggest that for figures 4, 6 7 and 8 that the authors use arrows and/or annotated text blocks to indicate precisely where the features are in the appropriate figures. This has been done effectively in figure 2, so the authors could use that as an example.
3. In my view the real strength of the paper is the use of gravity and seismic data for characterizing the sub sea structure of the coastal ocean transition zones. However, the authors do not provide reasons that would explain, for instance, the low gravity associated with the structure revealed by the seismic section. The authors have made some suggestions in section 4, but these should be amplified.

Reviewer 2 Report
In the introduction, the reason of choosing this region and the full explanation of the objectives and the question plan are not discussed. Please write them in details.
Place figure 2 above its caption. Give more details in the caption of the figure.
Edit line 79 and 238. The words (e.g. basin and length) are repeated twice. Such cases can be seen in different parts of the article.
Lines 82 and 83 are related to the acknowledgment section.
Please complete the captions of all figures and tables.
Add a section called “materials and methods” to the article.
Please revise the structure of the article. Consider the section titled “Results”. After it present the “discussion” section.
Revise lines 231 to 241.
In line 62 apply graben and half graben- like structures
In the figure 12, show also the volcanoes. You can discuss the relationship between the fault and volcano structures.
In my opinion, in Figure 15, the faults in the central part of the figure are not drawn correctly.
The conclusion does not need to be numbered.
Reviewer 3 Report
In connection with the reduction of noise caused by wave reflection(marine reflection seismic data), more information should be provided.
Initial migration velocity model need to described more.
The most important part of research is validity and accuracy, which is not tangible. Please explain.
Section 4 and Figure 12 regarding low and high gravity zone need separation and additional explanations.
Section 4 of conclusions should be revised (regarding other levels of data).
